# MRI Quantitative Evaluation of Muscle Fatty Infiltration

**Vito Chianca [1,2,\*], Bottino Vincenzo [2], Renato Cuocolo [3], Marcello Zappia [4,5,6], Salvatore Guarino [7], Francesco Di Pietto [8] and Filippo Del Grande [1]**

1 Radiology Department EOC IIMSI, 6900 Lugano, Switzerland
2 Ospedale Evangelico Betania, Via Argine 604, 80147 Naples, Italy
3 Department of Medicine, Surgery and Dentistry, University of Salerno, 84081 Baronissi, Italy
4 Department of Medicine and Health Sciences, University of Molise, 86100 Campobasso, Italy
5 Musculoskeletal Radiology Unit, Varelli Institute, 80126 Naples, Italy
6 Campolongo Hospital, 84121 Eboli, Italy
7 Department of Radiology, Azienda Ospedaliera dei Colli, Monaldi Hospital, 80131 Naples, Italy
8 Department of Radiology, Pineta Grande Hospital, 81030 Castel Volturno, Italy
\* Correspondence: vitochianca@gmail.com

**Abstract:** Magnetic resonance imaging (MRI) is the gold-standard technique for evaluating muscle fatty infiltration and muscle atrophy due to its high contrast resolution. It can differentiate muscular from adipose tissue accurately. MRI can also quantify the adipose content within muscle bellies with several sequences such as $T_1$-mapping, $T_2$-mapping, spectroscopy, Dixon, intra-voxel incoherent motion, and diffusion tensor imaging. The main fields of interest in musculoskeletal radiology for a quantitative MRI evaluation of muscular fatty infiltration include neuro-muscular disorders such as myopathies, and dystrophies. Sarcopenia is another important field in which the evaluation of the degree of muscular fat infiltration or muscular hypotrophy is required for a correct diagnosis. This review highlights several MRI techniques and sequences focusing on quantitative methods of assessing adipose tissue and muscle atrophy.

**Keywords:** MRI; DTI; $T_2$ mapping; radiomics; DWI

## 1. Introduction

In the last decade, quantitative imaging evaluation (QI) has been increasingly applied in radiological clinical practice. QI can correctly identify the pathological stages and better evaluate patients' follow-ups [1]. Magnetic resonance imaging (MRI) is the main imaging technique capable of giving quantitative information through specific sequences without ionizing radiation [2]. One of the main fields of interest for a quantitative evaluation in the musculoskeletal (MSK) system is fatty infiltration and muscle atrophy. QI evaluation quantifies muscular changes in several neuromuscular diseases such as muscular dystrophies and myopathies [3]. Moreover, with the average age increase in the world population, sarcopenia is another condition that requires a QI evaluation [4]. This is due to the direct connection with the prevalence and extent of the adipose infiltration of the locomotor muscles causing reduced muscle strength, mobility, and metabolic status changes in elderly patients [5,6]. For these reasons, it is crucial to non-invasively evaluate and monitor the state of adipose infiltration of the muscles. The measurements obtained through the use of optimized sequences allow for obtaining biomarker imaging of disease progression. In this way, it will be possible to carry out an earlier and more targeted therapy for patients.

This review article intends to overview the main quantitative MRI evaluation of adipose and muscular tissue focusing on the advantages, disadvantages, and limits of each sequence that radiologists can use during their clinical activity.

## 2. Chemical-Shift MR Imaging

The principle of chemical-shift imaging described by Dixon is based on different precession frequencies of fat and water protons at a specific magnetic field strength [7,8]. Two-point Dixon technique acquires two different images where proton spin magnetization vectors, at two different echo times (TE), are either in the same or in the opposite direction to each other. This acquisition allows the elaboration of in- and out-phase images [9]. Different signal intensities can then be used to generate water and fat images resulting in four different image contrasts (Figure 1). The Dixon technique provides an optimal homogeneous fat saturation and for this reason, it is preferred to other fat suppression techniques on a large field of view [10] or in areas with magnetic susceptibility for the presence of metallic components [11]. The main limitation of two-point Dixon techniques may be the $B_0$ heterogeneity, which causes the shift of fat and water peaks with the suppression of a wrong component; this is the so-called fat-water swapping effect that can be avoided by using unwrapping algorithms for the $B_0$ field heterogeneity compensation [9]. The three-point-Dixon technique is proposed to solve the limitation of the two-point-Dixon sequence. The presence of another echo acquisition with a different TE forms a complex with three equations about water content, fat content, and heterogeneity. This sequence shows a better signal-to-noise ratio (SNR) with a more homogeneous fat suppression and optimal spatial resolution [12]. Multi-echo Dixon techniques are also able to accurately quantify fat content in any voxel through fat-fraction maps where the grey value of each pixel is directly correlated with the fat infiltration [13]. Both two- or three-point Dixon sequences have been shown to quantify muscular fat fraction with excellent reliability in some anatomical areas such as rotator cuff muscles [14] or upper limbs after brachial plexus injuries [15]. Wieser et al. investigated 40 patients treated with arthroscopic rotator cuff repair with a 6-point Dixon sequence and reported a direct correlation between failed repair and the amount of fatty infiltration [16].

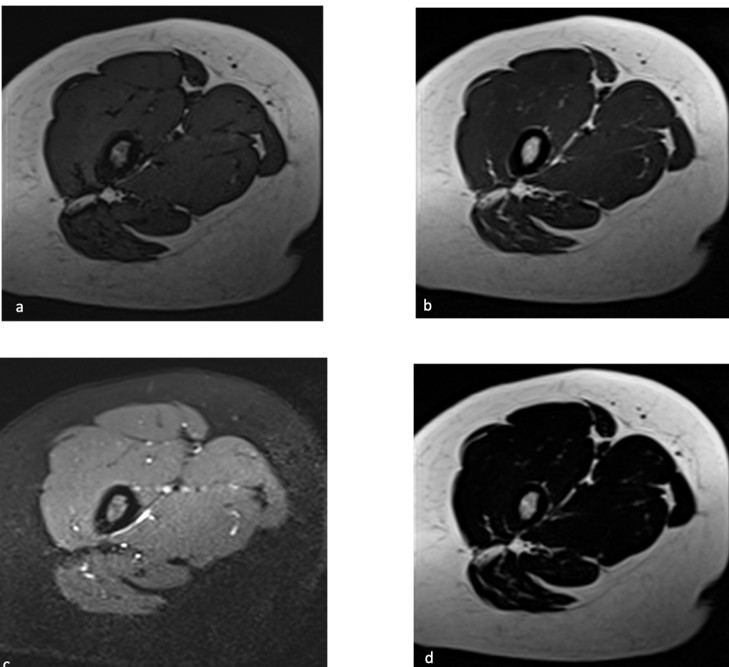

**Figure 1.** Dixon magnetic resonance sequence acquired on the thigh. Axial T1W out-phase (**a**), in-of-phase (**b**), 100% water images (**c**), and 100% fat images (**d**).

In the field of neuromuscular disorder, authors report that the three-point Dixon fat fraction shows a stronger correlation with a disease severity score than any other clinical tests in Duchenne muscular dystrophy patient's group [17]. Dahlqvist et al. proved that fat fraction derived by chemical-shift imaging of paraspinal muscles in facioscapulohumeral

dystrophy patients, correlated with clinical and genetic disease markers [18]. Another group used a three-point Dixon to quantify intramuscular fat fraction in Duchenne muscular dystrophy. The result shows a stronger correlation with a validated disease severity score than any other tested clinical examinations, such as isokinetic dynamometry of the knee extensor strength [17]. Another study reported higher accuracy and reliability of the three-point-Dixon sequence than conventional radiological methods for evaluating fat fractions for follow-up or therapeutic evaluation of Duchenne patients [19]. The 3D-multiple gradient-echo-Dixon sequence is reported to be a reproducible and sensitive technique able to highlight a significant difference in the fat fraction of thigh muscle, between Charcot–Marie–Tooth disease patients and volunteers [20].

### 3. MR Spectroscopy

MR spectroscopy (MRS) is a functional technique that provides biochemical information on human small metabolites according to their chemical-shift properties [21] on the basis of the MRS spectrum. MRS requires a high magnetic field to calculate small volumes spectra and at least 3 tesla (T) scanner is required for its higher signal-noise-ratio (SNR). Some studies reported that ultra-high magnetic field scanners (7 tesla) have advantages for MRS due to the improved SNR and the higher resolution [22,23]. Different techniques are reported for a qualitative and quantitative MRS evaluation; in particular, single-voxel or multi-voxel techniques can be used for proton MRS. Single-voxel spectroscopy (SVS) consists of the analysis of a single voxel of the selected region of interest (ROI) [24] while multi-voxel spectroscopy (MVS), is able to process simultaneously several voxels contained in a wider area [25]. Various metabolites can be differentiated depending on the MRS spectrum. In the setting of msk pathologies, phosphorus-31 (31P) MRS is used to detect metabolites containing phosphorus such as phosphocreatine, inorganic phosphate, adenosine triphosphate, and phosphocholine, which are overexpressed in case of energy consumption with related muscle changes [26,27]. Janssen et al., report that 31P-containing metabolite concentrations, measured with MRS, are strictly linked with muscle fat replacement and consequent muscle strength reduction in patients with facioscapulohumeral dystrophy [28]. However, MSK evaluation with 31P MRS requires specific hardware and software which limit its clinical use. Proton (1H) MRS does not require specialized hardware and can be easily added to a conventional MRI protocol [2]. 1H MRS can be used to measure intramuscular fatty infiltration, and some articles stated that it could represent a gold standard for noninvasive quantification of fat infiltration [29,30]. For this reason, MRS can be useful in the characterization and quantification of fat infiltration in chronic muscle pain [31], muscular dystrophies [32], or chronic myopathies [33].

The main disadvantage of the quantitative assessment of intramuscular lipid content with MRS is the significant sampling error due to the variability of the positioning of the volume of interest (VOI). Small changes in the VOI placement can determine a significant variability in fat quantification [30].

### 4. Relaxometry Mapping

$T_2$ mapping is a quantitative MR technique that measures the $T_2$ relaxation times of human tissues within a selected ROI [34,35]. From a technical point of view, this technique requires the acquisition of multiple images with different TEs (at least 3 different TEs) that allow the elaboration of a $T_2$ map [36,37]. Conventionally the $T_2$ map is reconstructed with fat saturation to avoid artifacts due to fatty infiltration that physiologically elevates $T_2$ values; so in case of fat infiltration evaluation, a $T_2$ map without fat suppression is mandatory [38] (Figures 2 and 3). Several acquisition methods have been proposed in the literature to evaluate $T_2$ values. In current clinical practice, multi-echo spin-echo sequences are typically used with various types of exponential fitting (mono-, bi-, or tri-exponential) for $T_2$ calculation [39]. However, these methods are sensitive to multiple confounding factors, such as $B_1$ inhomogeneities. Even if there are low differences in the protocol proposed by the different MR scanner vendors, the differences in the implementation of the

fitting process can be substantial. Some authors present a fast method for reconstructing $T_2$ maps obtained from different multi-echo spin-echo sequences and fit a method that gives reproducible results [39].

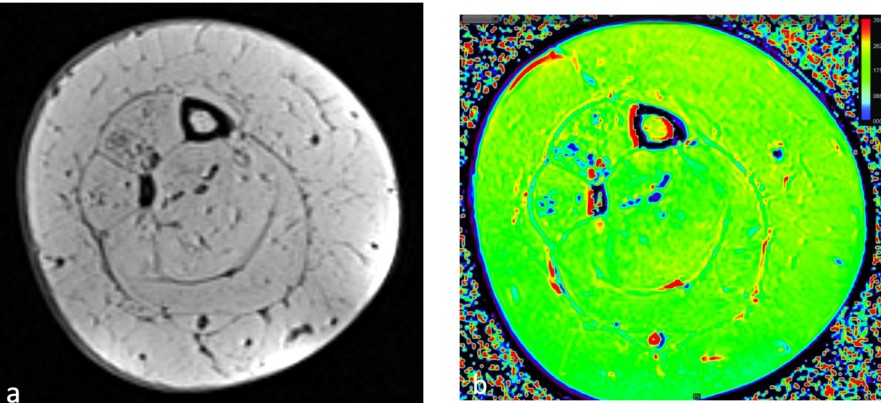

**Figure 2.** Axial T1w (**a**) image of a type III spinal muscular atrophy patient which shows massive fatty replacement of thigh muscles. Axial T2map image (**b**) shows very high $T_2$ relaxation time values of the corresponding muscle bellies. The mean T2map muscle value is 158 ms.

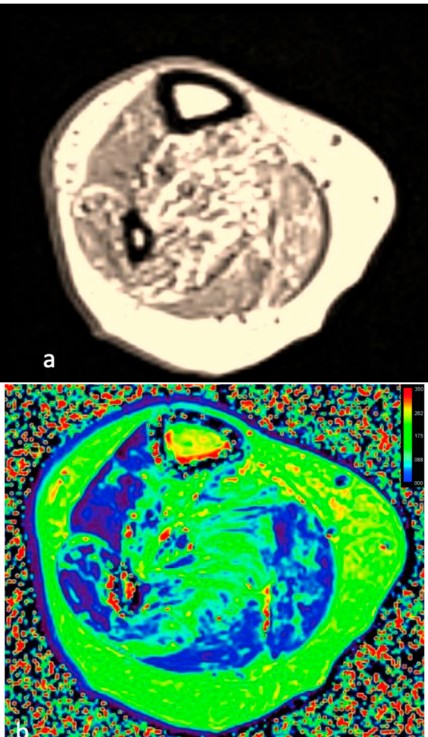

**Figure 3.** Axial T1w (**a**) and axial T2map (**b**) images of limb-girdle muscular dystrophy 2I patients show a partial fatty replacement of thigh muscles. The mean T2map value of muscles is 87 ms.

The correct interpretation of $T_2$ mapping in myopathies and dystrophies can be challenging because these patients often present edema and fatty infiltration, both conditions that elevate the $T_2$ values [40]. $T_2$ mapping remains a good support for a quantitative evaluation; however, in the literature, no defined cut-offs are described, so it is challenging to establish defined diagnostic criteria based on its values.

$T_1$ mapping is another method to quantify muscle fat fraction based on the modulation of $T_1$ values by the fat pool [41]. $T_1$ mapping imaging is often used to evaluate conditions related to oedema and fibrosis, where $T_1$ value changes reflect changes in water

mobility [42]. However, when fat is present in the voxel of interest, the resulting $T_1$ value is influenced by a complex of water and fat content. In the condition of fat replacement, the fat fraction amount is directly correlated with the decrease in longitudinal $T_1$ relaxation time [43]. Marty et al. evaluated ten healthy men and 30 men with Becker muscle dystrophy and reported high repeatability of the $T_1$ values measured with their high-resolution $T_1$ sequence able to discriminate healthy and dystrophic conditions in all the muscle groups (Student *t*-test, $p < 0.05$) [44].

*Diffusion-Weighted Imaging*

Diffusion tensor imaging (DTI) is an MRI-based technique evolved from the principles of DWI, which is mainly used in clinical practice for the evaluation of the central and peripheral nervous system [45–47]. DTI is based on the concept that water mobility in the human body does not move equally in all directions due to the presence of cell membranes muscle fibers or myelin sheath that restrict molecule diffusion. For this reason, water molecule diffusion is non-isotropic. For an accurate DTI sequence is mandatory to acquire DWI with high b values along at least 6 directions and a low b value DWI or a $T_2$-weighted sequence.

Higher b values increase gradient power and sequence diffusion weighting but reduce SNR. Authors reported an optimized sequence using 20–30 mm [3] voxel volumes, shortest TE, b values of 400–500 $s/mm^2$, and at least 10 gradient directions [48]. Although a clear benefit of using high b values is reported for central nervous system pathologies investigation, no articles report a definite clinical benefit of using high b values with ultra-high magnetic resonance field. About that point, Giraudo et al., use the same b value of 500 $s/mm^2$ on both 3 tesla and 7 tesla scanners [49]. Fractional anisotropy (FA) is the main parameter used to express fiber structural integrity; it ranges from 0 to 1; a value proximal to 1 expresses structural integrity while an FA value proximal to 0 expresses isotropic diffusion of molecules due to fiber damage [50].

Acquisition time depends mainly on the number of directions and magnetic field strength; there is a directly proportional relationship between scanning time and the number of directions, while there is an inversely proportional relationship between magnetic field strength and scanning time. Furthermore, the number of averages affects the scan time; the higher the number of averages, the longer the scan time will be. In the case of muscle evaluation 12 non-collinear directions are enough to obtain a satisfactory sequence [51]. Tractography is a DTI extension that allows 3-D visualization of muscle and nerve fibers [52,53]. Using MR with a higher magnetic field straight and higher magnetic gradients, nerve fascicle morphology and morphometric parameters can be better visualized [49].

It is possible to visualize the rarefaction of muscle fibers in the case of neurodegenerative diseases with consequent fat replacement. Klupp et al. compared paraspinal muscular DTI parameters and isokinetic dynamometer evaluation and reported a significant correlation between DTI measurements and functional tests [54].

In the setting of muscle dystrophies, Ponrartana et al. used DTI quantitative MRI parameters to evaluate Duchenne muscular dystrophy progression in the lower extremities and reported a high correlation between DTI values, and qualitative evaluation using the Mercuri grading [55]. Nevertheless, patients with a fat content of more than 45% within muscle bellies may determine an artificial decrease in ADC and a paradoxical artificial increase in FA values (Figure 4); for these reasons, progression of dystrophies positively correlates with FA and negatively with ADC values [56]. The main disadvantages of DTI are the complexity of the setup of the sequence, the scan times, and the need for a high magnetic field scanner with high-performance gradients.

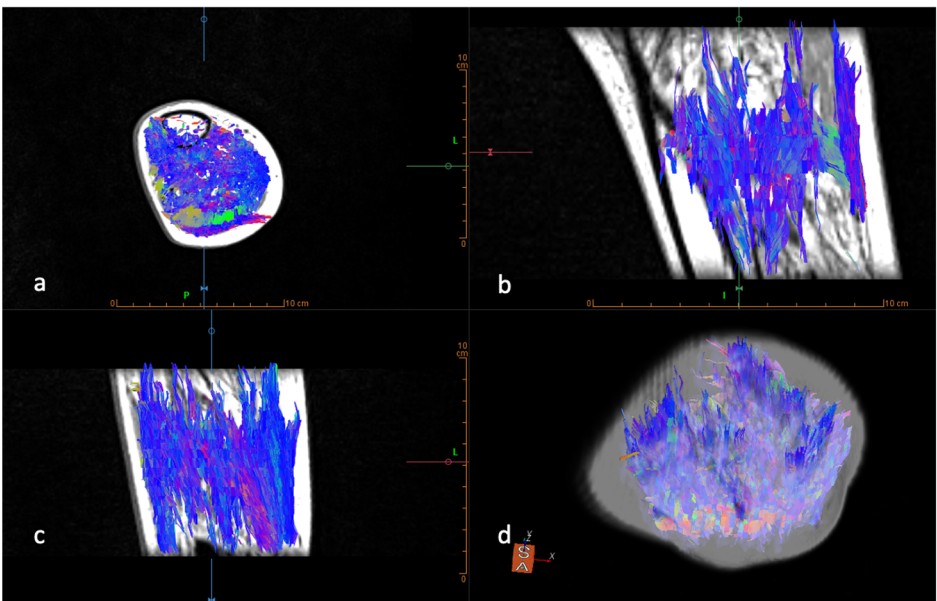

**Figure 4.** Axial (**a**), sagittal (**b**), coronal (**c**), and axial fusion tractography (**d**) images of a type II spinal muscular atrophy patient show rarefaction of regular muscle fibers. The DTI sequence (slice thickness: 3 mm, b value: 400, number of directions: 12) shows a mean FA value of 0.61 calculated on leg muscle compartments ROI.

Till now there were doubts about the reliability of the sequence on different scanners; Guggenberger et al. performed a wrist MR exam on 3 different 3-tesla scanners on 16 healthy volunteers and reported different FA and ADC values [57]. However, the comparison of the values with the standard deviation was small enough to not have an impact on a larger group. They also used different software for processing the values for each scanner. A recent study performed on 3 scanners (3 tesla) from different vendors, reported excellent results such as inter-vendor and inter-observer reliability using a single specific software [53].

The intravoxel incoherent motion (IVIM) is another imaging technique based on diffusion, that uses multiple b values to quantify the incoherently flowing vascular blood pool signal from that of the tissue diffusion signal [58]. IVIM reflects the water molecule's microscopic motion in each MRI voxel of the intracellular space, extracellular space, and blood capillaries [59]. With this imaging technique, diffusion and perfusion are affected by tissue characteristics such as the presence of cellular membranes, the fluid viscosity, and the velocity of perfusing spins [60]. One of the limitations of the use of this imaging method is the potential degradation of images caused by cardiac activity, and motion artifacts which limited, for years, the technique to the study of the central nervous system only [61]. However promising results were reported about the IVIM in the msk system, particularly in the differentiation of myositis from muscular dystrophy [62] or for quantification of muscle activation [63].

## 5. Artificial Intelligence

The use of artificial intelligence (AI) in the medical field is progressively revolutionizing the daily clinical life of physicians. AI provides an interesting tool that improves performance in all diagnostic fields [64]. AI is a field of computer science dedicated to the elaboration of systems tasks that generally require human intelligence [65]. Different subfields of AI are proposed, but the most investigated for future development are machine learning (ML) and deep learning (DL) [66]. ML trained and tested different algorithms to recognize specific image characteristics by learning processes from an origin dataset. DL is another subtype of AI based on the use of different layers of neural networks (NN) that allow a computer to analyze a large amount of data and identify useful features for a correct classification [67]. Although most AI studies in the msk field focus on tumor recognition

and classification [68–71], several authors investigated the use of AI in the evaluation of muscle fat infiltration. They reported accurate and rapid, quantitative assessment [72]. Ro et al. tested NN to automatically evaluate the occupation ratio and fatty infiltration of rotator cuff muscles on 240 patients who underwent shoulder MRI. They reported a strong negative correlation between the occupation ratio via convolutional NN and fatty infiltration via the Otsu thresholding method [73]. Regarding muscle dystrophies, Verdú-Díaz et al. collected 976 pelvic and lower limbs muscle MRIs from 10 different muscle dystrophies. They tested ML capabilities in detecting and quantifying abnormalities by comparing them with those of four specialists. The best model tested showed 95.7% accuracy, with 92.1% sensitivity and 99.4% specificity higher than the values the specialists showed [74]. Ding et al. tested AI for automated thigh muscle segmentation showing excellent accuracy and higher reproducibility in fat fraction quantification compared to manual segmentation [75].

Furthermore, ML and DL in particular are also revolutionizing other aspects of MRI in this domain, especially image acquisition and reconstruction [76]. In particular, techniques based on Generative Adversarial Networks and model-based image reconstruction from raw k space data have proven valuable to the lower acquisition time of quantitative imaging data or improve the robustness of the resulting parametric maps. As these represent two of the main current limitations for widespread clinical adoption, such technical developments may greatly aid in the clinical implementation of quantitative MRI assessment of muscle fatty infiltration.

Further research will be necessary for the clinical validation of these AI tools.

## 6. Conclusions

Adipose infiltration and adipose involution are important findings of inflammatory-degenerative muscle pathologies. When present, these changes indicate an advanced state of disease. MRI has multiple techniques and sequences for both qualitative and quantitative assessment such as chemical-shift sequences, spectroscopy, relaxometry, diffusion tensor imaging, and IVIM. All these sequences can calculate the degree of fat infiltration differently as shown in Table 1. However, each sequence shows some limitations that prevent their routine use in clinical practice. However, the implementation of study protocols with at least one of these sequences is necessary to obtain a tailored therapy for patients. The development of AI will allow a reduction of motion artifacts and allow to perform faster post-processing to obtain the quantitative values. For all these reasons, in modern medicine, is crucial for a radiologist to know methods for quantifying muscle fat content.

**Table 1.** Summary of different MR imaging techniques.

| Imaging Techniques | Type of Evaluation | Disadvantages |
|---|---|---|
| Dixon | Quantification of muscular fat fraction with excellent reliability. | $B_0$ heterogeneity, which causes the shift of fat and water peaks with the suppression of a wrong component. |
| Spectroscopy | MRS evaluates metabolic muscle changes in case of muscle fat infiltration through the quantitative analysis of metabolites containing phosphorus. | Significant sampling error due to the variability of the positioning of the volume of interest. |
| Relaxometry | Quantitative relaxation time evaluation of the selected muscle. | The presence of edema determines errors in quantifying muscular fat infiltration. |
| DTI | Quantitative evaluation of the degree of muscular adipose infiltration by calculating the fraction anisotropy. | Complexity of the sequence setup and the scan times. |
| IVIM | Quantitative evaluation of the incoherently flowing vascular blood signal from that of the other tissue. | Cardiac activity and motion artifacts. |

**Author Contributions:** Writing, V.C. and S.G.; formal preparation, B.V.; review and editing, R.C., M.Z., F.D.P. and F.D.G. All authors have read and agreed to the published version of the manuscript.

**Funding:** This research received no external funding.

**Institutional Review Board Statement:** Not applicable.

**Informed Consent Statement:** Not applicable.

**Data Availability Statement:** Not applicable.

**Conflicts of Interest:** The authors declare no conflict of interest.

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
