# Peer review of "MRI Quantitative Evaluation of Muscle Fatty Infiltration"

_magnetochemistry, doi:10.3390/magnetochemistry9040111_

Round 1

Reviewer 1 Report

This is a review about the application of different quantitative MR imaging techniques on Musculo-skeletal diseases, specifically for the characterization of fatty infiltration.

This specific topic seems to be missing in the literature, although some reviews about quantitative MRI for muscle characterization are present. However, I believe that the manuscript is quite generic and superficial in the description of the different quantitative techniques. Significant modifications should be implemented to improve its value. Specifically:

1.       I suggest the authors to include a section about relaxometry mapping in general (not only T2 mapping), since several works used e.g. T1 mapping to characterize fat-muscle tissues (see for example https://doi.org/10.1007/s00330-018-5433-z, https://doi.org/10.1002/mrm.26113, https://doi.org/10.1186/s12891-022-05640-y)

2.       In general, I found that sections about MRI techniques were poor from the modeling point of view. I think that a deeper description of the physical model behind the estimation of parametric maps would help the reader in better understanding the information and limitations related to specific acquisitions.

3.       Beside the previous comment, I also suggest to describe the methods adopted to fit the models and obtain parametric maps (e.g. https://doi.org/10.3389/fneur.2021.630387). A critical discussion about issues and limitations related to estimation errors would be very interesting. In addition, the new frontier for quantitative MRI mapping estimation is the adoption of deep learning techniques (e.g. https://doi.org/10.1002/NBM.4416, https://doi.org/10.1002/mrm.29128, https://doi.org/10.1002/nbm.4774, https://doi.org/10.1016/j.neuroimage.2020.117017), that should be mentioned.

4.       In section 5, DTI is described and discussed. However, since the discussion includes also indices such as ADC, which does not provide information about tractography, I suggest to rename this section with a more general “Diffusion MRI”. In this case, I also suggest to include other dMRI models, such as IntraVoxel Incoherent Motion, that has been adopted to characterize muscle microstructure (https://doi.org/10.1002/jmri.27875, https://doi.org/10.1002/nbm.3922), also in muscular dystrophies (https://doi.org/10.1016/j.acra.2020.04.022).

Minor issues:

5.       In Figure 2 and 3, colorbar should be included

6.       Section 6 is named as “Texture analysis”; however, texture analysis was never mentioned. If the intention was to include radiomics studies, the authors should better highlight them. Otherwise, I suggest to change the title to introduce the general concept of AI.

7.       Schematic tables able to provide a synthetic representation of the main results found in literature using the described imaging techniques would be very helpful for the readers.

8.       Introduction should better describe the motivation and originality of this review

9.       The conclusion should be extended with current limitations and possible future expansions in the field.

10.       Some typos are present in the manuscript. Please, carefully check English language and grammar.

Author Response

Thank you for the interest suggestions.

This is a review about the application of different quantitative MR imaging techniques on Musculo-skeletal diseases, specifically for the characterization of fatty infiltration.

This specific topic seems to be missing in the literature, although some reviews about quantitative MRI for muscle characterization are present. However, I believe that the manuscript is quite generic and superficial in the description of the different quantitative techniques. Significant modifications should be implemented to improve its value. Specifically:

  1. I suggest the authors to include a section about relaxometry mapping in general (not only T2 mapping), since several works used e.g. T1 mapping to characterize fat-muscle tissues (see for example https://doi.org/10.1007/s00330-018-5433-z, https://doi.org/10.1002/mrm.26113, https://doi.org/10.1186/s12891-022-05640-y)

Answer: Thank you for your suggestion. We change the name of the paragraphi in relaxometry mapping and add a section regarding T1 mapping.

  1. In general, I found that sections about MRI techniques were poor from the modeling point of view. I think that a deeper description of the physical model behind the estimation of parametric maps would help the reader in better understanding the information and limitations related to specific acquisitions.
  2. Beside the previous comment, I also suggest to describe the methods adopted to fit the models and obtain parametric maps (e.g. https://doi.org/10.3389/fneur.2021.630387). A critical discussion about issues and limitations related to estimation errors would be very interesting. In addition, the new frontier for quantitative MRI mapping estimation is the adoption of deep learning techniques (e.g. https://doi.org/10.1002/NBM.4416, https://doi.org/10.1002/mrm.29128, https://doi.org/10.1002/nbm.4774, https://doi.org/10.1016/j.neuroimage.2020.117017), that should be mentioned.

Answer 2-3: Thank you for your suggestion. We add a deep description of MRI techniques, in particular of T2 mapping. We add the limitation of this sequence. We add a section regarding the article https://doi.org/10.3389/fneur.2021.630387 and also we add in the AI paragraph section regarding deep learning techniques.

  1. In section 5, DTI is described and discussed. However, since the discussion includes also indices such as ADC, which does not provide information about tractography, I suggest to rename this section with a more general “Diffusion MRI”. In this case, I also suggest to include other dMRI models, such as IntraVoxel Incoherent Motion, that has been adopted to characterize muscle microstructure (https://doi.org/10.1002/jmri.27875, https://doi.org/10.1002/nbm.3922), also in muscular dystrophies (https://doi.org/10.1016/j.acra.2020.04.022).

Anser 4: Thank you for the suggestion. We add a section regarding IVIM in the text.

Minor issues:

  1. In Figure 2 and 3, colorbar should be included

Answer: we add a colorbar in the figures.

  1. Section 6 is named as “Texture analysis”; however, texture analysis was never mentioned. If the intention was to include radiomics studies, the authors should better highlight them. Otherwise, I suggest to change the title to introduce the general concept of AI.

6: Thank you for your suggestion. We renamed the paragraph with the name Artificial intellingence.

  1. Schematic tables able to provide a synthetic representation of the main results found in literature using the described imaging techniques would be very helpful for the readers.

Answer: We add a table regarding the different imaging techniques.

  1. Introduction should better describe the motivation and originality of this review

Answer: Thank you for your suggestion. We have implemented the introduction.

  1. The conclusion should be extended with current limitations and possible future expansions in the field.

Answer: Thank you for your suggestion. We have implemented the conclusion.

  1. Some typos are present in the manuscript. Please, carefully check English language and grammar.

Answer: We have corrected some English errors.

Reviewer 2 Report

I would like to thank the authors for this submission. The work is interesting but needs to be improved before it is suitable for publication.

Abstract: Abstract it brief, I recommend the authors to expand this part.

 Page 2, line 69: The source citation is missing.

 Page 3, line 83. Did any of the authors use a higher magnetic field than 3T for MR spectroscopy of muscles?

 Page 4, line 128-130. DTI is regularly used in clinical practise to assess the CNS; however, I do not believe that this imaging technique has been used for anything more than research purposes in the PNS. If so, please add the reference(s).

 Page 4, lines 134-135. what about stronger magnetic gradients? Are these always required to achieve higher b-values in musculoskeletal MRI? Please comment.

 Page 4, line 139: What about other parameters, e.g. FOV, number of averages? Do they affect the time frame?

 Page 4, lines 139-141: This reference (41) needs more thorough discussion.

 Page 4, line 143: There is a missing reference to a study that uses tractographic representations for muscles. Please add.

 Page 5, lines 157-59. Some authors claim otherwise. Ref.: doi:10.1016/j.ejrad.2013.05.011

 Page 4, line 143: Using MR with higher magnetic gradients, even nerve fascicle morphology and morfometric parameters can be accurately visualised.

Fig. 4. the mean FA of what? What was the ROI? How was it tracked, what direction, slice thickness?

Line 230. Can you please explain how MR is able to detect muscle fibre type?

Author Response

Thank you for the important suggestions.

I would like to thank the authors for this submission. The work is interesting but needs to be improved before it is suitable for publication.

Abstract: Abstract it brief, I recommend the authors to expand this part.

Answer: Thank you for your suggestion. We have implemented the abstract.

 Page 2, line 69: The source citation is missing.

Answer: We added the citation.

 Page 3, line 83. Did any of the authors use a higher magnetic field than 3T for MR spectroscopy of muscles?

Answer: Thank you for your suggestion. We added two references about 7 tesla MRI spectroscopy evaluation of muscle.

 Page 4, line 128-130. DTI is regularly used in clinical practise to assess the CNS; however, I do not believe that this imaging technique has been used for anything more than research purposes in the PNS. If so, please add the reference(s).

Answer: We partially agree with his suggestion.

In selected cases we use DTI in clinical protocols as reported in the. Reference 49.

However, they alone are not enough for a definitive diagnosis. We add another references of the use of DTI in PNS.

 Page 4, lines 134-135. what about stronger magnetic gradients? Are these always required to achieve higher b-values in musculoskeletal MRI? Please comment.

Answer: We add some technical aspects of the sequences.

 Page 4, line 139: What about other parameters, e.g. FOV, number of averages? Do they affect the time frame? And Page 4, lines 139-141: This reference (41) needs more thorough discussion.

Answer: We add some technical aspects of the sequences.

 Page 4, line 143: There is a missing reference to a study that uses tractographic representations for muscles. Please add.

Answer: We add the reference.  

 Page 5, lines 157-59. Some authors claim otherwise. Ref.: doi:10.1016/j.ejrad.2013.05.011

Answer: Thank you for your suggestion. We have added this reference and explained the differences

 Page 4, line 143: Using MR with higher magnetic gradients, even nerve fascicle morphology and morfometric parameters can be accurately visualised.

Answer: Thank you for your suggestion. We add this consideration.

Fig. 4. the mean FA of what? What was the ROI? How was it tracked, what direction, slice thickness?

Answer: The DTI sequence (slice thickness: 3 mm, b value: 400, number of directions: 12) shows mean FA value of 0.61 calculated on leg muscle compartments ROI.

Line 230. Can you please explain how MR is able to detect muscle fibre type?

Answer: Thank you for your suggestion. We specify how it can be possible.

Round 2

Reviewer 1 Report

The authors have only partially revised the manuscript, since part of my comments were not taken into account. Although the paper has been improved from some points of view, I believe that the current version is not ready for publication yet.

Specifically:

1.       Points 2 and 3 of my previous revision, about the description of the mathematical and physical model behind the acquisitions and the methods adopted to fit them, including deep learning strategies, were only partially addressed. Deep learning is currently treated as a technique to perform classification starting directly from images. My request, instead, was to add deep learning as a powerful technique to improve model fitting to obtain parametric maps (e.g. https://doi.org/10.1002/NBM.4416, https://doi.org/10.1002/mrm.29128, https://doi.org/10.1002/nbm.4774, https://doi.org/10.1016/j.neuroimage.2020.117017). This point is still missing.

2.       The authors stated in their reply to point 7 that they added a schematic table about the different imaging techniques. However, I didn’t find it in the revised version of the manuscript.

3.       Section 5 should be renamed in “Diffusion-weighted imaging” rather than “Diffusion tensor imaging”, since not only DTI is reported there.

Author Response

We thank the Reviewer for further clarifying this point. The "Artificial Intelligence" section of the text has now been revised accordingly, including an explicit mention of the potential and value of DL-based image reconstruction and quantitative map calculation techniques in the context of MRI assessment of muscle fatty infiltration.

We add the table.

Reviewer 2 Report

The authors satisfactorily addressed all my comments.

Author Response

Thank you for your comment.